# Can We Estimate Functionality of Soil Microbial Communities from Structure-Derived Predictions? A Reality Test in Agricultural Soils

Claudia Breitkreuz,[a] Anna Heintz-Buschart,[a,b] François Buscot,[a,b] Sara Fareed Mohamed Wahdan,[a] Mika Tarkka,[a,b] Thomas Reitz[a,b]

aDepartment of Soil Ecology, UFZ—Helmholtz Centre for Environmental Research, Halle, Germany
bGerman Centre for Integrative Biodiversity Research (iDiv) Halle-Jena-Leipzig, Leipzig, Germany

**ABSTRACT** Computational approaches that link bacterial 16S rRNA gene amplicon data to functional genes based on prokaryotic reference genomes have emerged. This study aims to validate or refute the applicability of the functional gene prediction tools for assessment and comparison of community functionality among experimental treatments, inducing either fast or slow responses in rhizosphere microbial community composition and function. Rhizosphere samples of wheat and barley were collected in two consecutive years at active and mature growth phases from organic and conventional farming plots with ambient or future-climate treatments of the Global Change Experimental Facility. Bacterial community composition was determined by 16S rRNA gene amplicon sequencing, and the activities of five extracellular enzymes involved in carbon ($\beta$-glucosidases, cellobiohydrolase, and xylosidase), nitrogen (N-acetylglucosaminidase), and phosphorus (acid phosphatase) cycles were determined. Structural community data were used to predict functional patterns of the rhizosphere communities using Tax4Fun and PanFP. Subsequently, the predictions were compared with the measured activities. Despite the fact that different treatments mainly drove either community composition (plant growth phase) or measured enzyme activities (farming system), the predictions mirrored patterns in the treatments in a qualitative but not quantitative way. Most of the discrepancies between measured and predicted values resulted from plant growth stages (fast community response), followed by farming management and climate (slower community response). Thus, our results suggest the applicability of the prediction tools for comparative investigations of soil community functionality in less-dynamic environmental systems.

**IMPORTANCE** Linking soil microbial community structure to its functionality, which is important for maintaining health and services of an ecosystem, is still challenging. Besides great advances in structural community analysis, functional equivalents, such as metagenomics and metatranscriptomics, are still time and cost intensive. Recent computational approaches (Tax4Fun and PanFP) aim to predict functions from structural community data based on reference genomes. Although the usability of these tools has been confirmed with metagenomic data, a comparison between predicted and measured functions is so far missing. Thus, this study comprises an expansive reality test on the performance of these tools under different environmental conditions, including relevant global change factors (land use and climate). The work provides a valuable validation of the applicability of the prediction tools for comparison of soil community functions across different sufficiently established soil ecosystems and suggest their usability to unravel the broad spectrum of functions provided by a given community structure.

Address correspondence to Claudia Breitkreuz, claudia.breitkreuz@ufz.de, or Anna Heintz-Buschart, anna.heintz-buschart@idiv.de.

**KEYWORDS** GCEF, PanFP, Tax4Fun, agriculture, bacterial communities, barley, climate change, enzymes, wheat

Over the last decades, we experienced a rapid advancement of molecular approaches to explore structural diversity of soil microbial communities. The use of next-generation amplicon sequencing allows for high-resolution analyses of microbial community structure, e.g., on its temporal dynamics and adaptation to different environmental conditions (1, 2). Corresponding studies revealed that soil microbial communities change over the growing season (3, 4) and are dependent on the plant species (5–8) as well as on the plant development stage (8–11). Moreover, soil type as well as land-use- and management-related variations in pH and available nutrient concentrations shape soil microbial communities (12–15). However, it remains challenging to determine the functional traits of a given microbial community in order to estimate resultant soil processes and ecosystem services (16). This is because soil processes and functions can be maintained in spite of community shifts by functional redundancy (reviewed in reference 17), while others may be lost by losing individual, possibly even low-abundant, key species (18). Thus, it is crucial to have information on the traits of all present taxa to derive the functionality of the whole community (19–21).

Cultivation has been traditionally used to cross-examine the taxonomic and functional properties of bacteria. Even though the longstanding "1% cultivability paradigm" has been questioned in recent discussions (22, 23), cultivation-based approaches are hardly meaningful for functional trait assessment in environmental samples, since trait variation is strongly reduced by studying only few isolates (24). When measuring aggregated functional properties of the microbial community, e.g., by analyzing community enzyme activities (25–27) or gene expression profiles (28, 29), it is often difficult to assign activities to certain taxa. Available methods that link structural and functional information of bacteria include stable isotope labeling of substrates and subsequent amplicon sequencing of isotope-enriched DNA or RNA (28, 30) or using genome-resolved metagenomics or metatranscriptomics (31, 32). Nevertheless, capturing the functional diversity of whole microbial communities in depth and breadth with these methods remains cost and time intensive (32).

Computational prediction tools in microbial ecology, such as Tax4Fun and PanFP, offer the possibility to translate structural community data into ecosystem functions in a cost-effective way (33–35). These approaches use the link between bacterial 16S rRNA gene amplicon sequencing and functional gene annotations of prokaryotic reference genomes. As output, the programs provide abundance estimates of functional genes. The applicability of both tools has been validated by comparison of the predicted functional gene abundance with the number of detected genes in the respective metagenome (36–38). Median Spearman rank correlation coefficients range up to 0.87 for Tax4Fun (36) and 0.80 for PanFP (37), suggesting good approximations of functional profiles. At the same time, assessment of whether and how well such predicted functional profiles mirror microbial community trait expression and thus allow estimation of ecosystem processes is still missing.

In this study, the activity potentials of five extracellular microbial enzymes ($\beta$-glucosidase, cellobiohydrolase, xylosidase, $N$-acetylglucosaminidase, and acid phosphatase) were measured and compared to the abundances of the respective genes predicted with PanFP and Tax4Fun based on Illumina MiSeq amplicon sequencing data. These enzymes were chosen because (i) they play a crucial role in soil C, N, and P cycling, (ii) their activities are commonly measured in environmental studies as representative of soil function, and (iii) the protocols and assays for activity determination are standardized and well established.

We expected that a linear link might not be conceivable, since a direct correlation of gene abundance and its related function would require that (i) the genes of interest are constitutively transcribed to mRNA, (ii) the mRNAs are translated into proteins, (iii) all proteins responsible for the same reaction have the same kinetics and optimal conditions for

activity, and (iv) all enzymes have the same life span (28). In reality, gene expression and enzyme secretion are not consistent but are regulated in response to soil conditions. Moreover, the life span of extracellular enzymes in soil can range from hours to months, depending on local biotic and abiotic soil parameters (28, 39). Thus, enzyme activity measurements depict the situation in soil at a certain point in time but do not necessarily reflect short-term changes in microbial community composition (39, 40). Nevertheless, we assumed that patterns of measured enzyme activities follow those of the corresponding functional gene abundances in the microbial communities along treatments or environmental gradients that exert a continuous and steady impact.

This study aimed to evaluate this assumption and to validate or refute the applicability of the functional gene prediction tools for assessment and comparison of soil processes. Since the functional predictions by Tax4Fun and PanFP refer exclusively to bacterial genomes, we selected croplands as study systems. Agricultural soils are usually dominated by bacteria, whereby the fungal contribution to enzyme profiles is minimized (41, 42). We collected rhizosphere soils, i.e., the hot spot for abundance, activity, and turnover of soil bacteria (43), of wheat (and barley in the subsequent year) from agricultural plots of the Global Change Experimental Facility (GCEF) (44). This experimental field platform cross-manipulates climatic conditions (ambient versus future) and farming management (conventional versus organic farming). Both experimental treatments are known to steer structure and function of bacterial communities (12–14). The small but continuous impact of changed climatic conditions induces a slow response of the soil community, whereas the adaptation to different management measures induces quicker community responses. To account for very rapid responses, we collected rhizosphere samples at two different plant growth phases: active biomass production and mature phase. The dynamics of roots from active to mature growth stages (45, 46) are known to cause rapid temporal changes in rhizobacterial community structures (reviewed in reference 47). From all collected samples, we determined the rhizobacterial community composition using 16S rRNA gene amplicon sequencing, estimated the functional gene abundances by the prediction tools Tax4Fun and PanFP, and measured the enzyme activity potentials.

We hypothesized that (i) deviations between the predicted traits and the measured enzyme activities show a positive correlation with the speed of the community's response to the treatments. Thus, the strongest deviations should be related to plant growth phases (strong dynamics, rapid adaptation) followed by the impact of the farming management, while the most concurrent patterns should be observed along the climate treatments (slow, consistent community adaptation). We further hypothesized that (ii) across growth phases, the deviations are more pronounced during the mature growth phase. Plants stimulate rhizobacterial growth and activity by a gradually increasing release of rhizodeposits during active growth but strongly reduce rhizodeposition when reaching maturity, inducing a reduction of bacterial biomass (9). Accordingly, functional gene abundance drops quickly, while there is a delay for enzyme activity. We also hypothesized that (iii) deviations in measured and predicted values are more pronounced in conventional farming soil, as these systems experience more disturbances by, e.g., pesticide application, compared to that of organic soil. Finally, we hypothesized that (iv) under future climatic conditions, with larger variability of annual precipitation, the deviations between measured and predicted functions are more pronounced than under ambient climatic conditions.

## RESULTS

**Experimental treatments drive rhizosphere community composition and enzyme activities. (i) Impact on the bacterial community composition.** The effect of the experimental treatments and the related differences in abiotic soil parameters (for more information on edaphic parameters and impact on community composition refer to Material S1 and S2, respectively, in the supplemental material) on the rhizobacterial community composition was studied for both crops in the two consecutive years of cultivation (Fig. 1). In the first year, when wheat was cultivated, growth phase was the main driver for bacterial community composition (permutational multivariate analysis

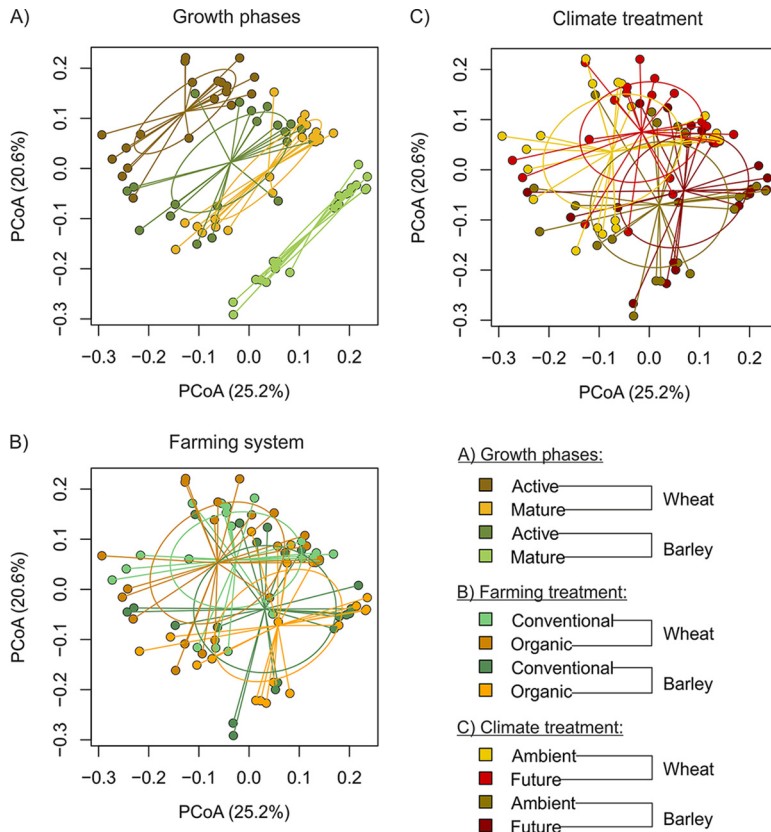

**FIG 1** Principal coordinates analysis for beta-diversity of bacterial rhizosphere communities. The points are colored and circled according to growth phases (A), farming system (B), and climate treatment (C).

of variance [PERMANOVA], $R^2 = 0.26$, $P < 0.001$) (Fig. 1A), followed by farming system (PERMANOVA, $R^2 = 0.08$, $P = 0.003$) (Fig. 1B) and climate treatment (PERMANOVA, $R^2 = 0.03$, $P = 0.18$) (Fig. 1C). In a comparable way, rhizobacterial community composition of barley in the subsequent year was affected in decreasing order by growth phase (PERMANOVA, $R^2 = 0.22$, $P < 0.001$) (Fig. 1A), farming system (PERMANOVA, $R^2 = 0.04$, $P = 0.08$) (Fig. 1B), and climate treatment (PERMANOVA, $R^2 = 0.03$, $P = 0.33$) (Fig. 1C). In line with these results, analysis and visualization of indicator species in a bipartite network indicated a strong grouping of species according to the growth phase and farming system in the wheat rhizosphere as well as according to the growth phase in the barley rhizosphere (Material S3). Besides, wheat and barley strongly differed in their rhizobacterial community composition (PERMANOVA, $R^2 = 0.13$, $P < 0.001$).

**(ii) Impact on rhizosphere enzyme activities.** Farming system and the related differences in edaphic parameters (Material S1) were the main drivers of enzyme activities (Table 1; Material S2). Thereby, higher enzyme activities were found in rhizosphere soil from conventional farming than in the ones from organic farming, which was evident for wheat at both growth phases, while for barley, it was mainly observed in the active growth phase (see the blue boxes in Fig. 2). The effects of the growth phase and of the climate treatment on rhizosphere enzyme activities were comparably weak, with significant impacts of individual extracellular enzymes and in a crop-specific manner (Table 1). The growth phase affected chitinase activity in wheat (active < mature) (Fig. 2C) and acid phosphatase activity in barley rhizosphere (active > mature) (Fig. 2D) (Table 1). Climate treatment effects were found for the activities of xylosidases and acid phosphatases in the rhizosphere of wheat and for the activity of cellulases in the rhizosphere of barley (Table 1). Besides, all enzyme activities strongly differed ($P < 0.001$) between wheat and barley rhizospheres, with higher enzyme activities in the wheat rhizosphere (Fig. 2).

**TABLE 1** Drivers of rhizosphere enzyme activities[a]

| Enzyme | P value[b] | | | |
|---|---|---|---|---|
| | Farming | Growth phase | Climate | Growth phase × farming |
| Wheat | | | | |
| Glucosidases | <0.001 | 0.76 | 0.11 | 0.57 |
| Xylosidases | 0.007 | 0.09 | 0.04 | 0.38 |
| Chitinases | <0.001 | 0.002 | 0.41 | 0.19 |
| Phosphatases | <0.001 | 0.19 | 0.04 | 0.21 |
| Cellulases | <0.001 | 0.84 | 0.77 | 0.91 |
| Barley | | | | |
| Glucosidases | <0.001 | 0.71 | 0.32 | 0.007 |
| Xylosidases | <0.001 | 0.15 | 0.67 | 0.03 |
| Chitinases | 0.02 | 0.24 | 0.19 | 0.14 |
| Phosphatases | <0.001 | <0.001 | 0.79 | <0.001 |
| Cellulases | <0.001 | 0.85 | 0.04 | 0.05 |

[a]Activities of β-glucosidases, xylosidases, N-acetylglucosaminidases (chitinases), acid phosphatases, and cellobiohydrolases (cellulases) were tested against the factors farming system, growth phase, climate, and interaction of farming system and growth phase.
[b]Significant impacts according to ANOVA are indicated by italic font.

**Patterns of measured enzyme activities compared to predicted enzyme gene abundances in the rhizosphere. (i) Correlations along and relative changes between factors of growth phase, farming system, and climate.** Spearman rank correlations were tested to identify common and specific patterns of predicted functional gene abundances and measured enzyme activities over both growth phases ($n = 40$) (Table S1). In the wheat rhizosphere, a positive correlation between measured and Tax4Fun- as well as PanFP-predicted values was found for xylosidases and with a trend observed for acid phosphatases. Contrary to that, in the rhizosphere of barley, functional gene abundances predicted by Tax4Fun were found to be positively correlated with the measured enzyme activities of glucosidases, xylosidases, chitinases, and cellulases. Regarding PanFP predictions, significant correlations with the measured activities were indicated for xylosidases and chitinases and with a trend observed also for glucosidases and cellulases.

Assessing the two growth phases separately, the significance level of correlations was commonly reduced, mainly due to the lower number of samples ($n = 20$). Nevertheless, we

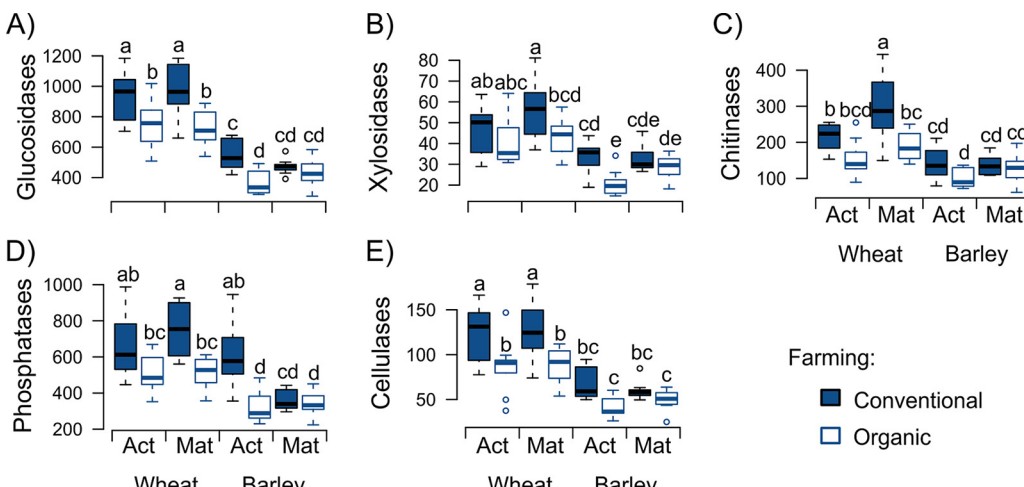

**FIG 2** Impacts of farming system, crop species, and crop growth phase on measured enzyme activities (nmol g soil$^{-1}$ h$^{-1}$). β-glucosidases (A), xylosidases (B), N-acetylglucosaminidases (chitinases) (C), acid phosphatases (D), and cellobiohydrolases (cellulases) (E). Measured enzyme activities at the active (Act) and mature (Mat) growth phases in conventional and organic farming soils are given. Different lowercase letters within each panel indicate significant differences between the treatments ($P < 0.05$) according to Tukey's HSD.

Microbiology Spectrum

**TABLE 2** Relative changes between factors of growth phase (active versus mature), farming system (conventional versus organic), and climate (ambient versus future) treatment[a]

| | Relative change | | | | | | | | |
| | Active vs mature | | | Conventional vs organic | | | Ambient vs future | | |
| Enzyme | Tax | Pan | Enzymes | Tax | Pan | Enzymes | Tax | Pan | Enzymes |
|---|---|---|---|---|---|---|---|---|---|
| Wheat | | | | | | | | | |
| Glucosidase | 0.27 | 0.53 | 0.02 | −0.23 | −0.23 | −0.23 | −0.20 | −0.18 | −0.08 |
| Xylosidase | 0.52 | 0.55 | 0.14 | −0.25 | −0.24 | −0.19 | −0.24 | −0.18 | −0.14 |
| Chitinase | 0.15 | 0.36 | 0.31 | −0.27 | −0.23 | −0.32 | −0.17 | −0.21 | −0.07 |
| Phosphatase | 0.89 | 0.59 | 0.09 | −0.26 | −0.25 | −0.28 | −0.27 | −0.22 | −0.13 |
| Cellulase | 0.84 | 0.40 | 0.02 | −0.22 | 0.02 | −0.31 | −0.30 | 0.01 | −0.02 |
| Barley | | | | | | | | | |
| Glucosidase | 0.60 | 1.15 | −0.02 | −0.28 | −0.22 | −0.22 | 0.00 | 0.02 | 0.06 |
| Xylosidase | 0.73 | 1.30 | 0.12 | −0.26 | −0.19 | −0.25 | −0.03 | 0.01 | 0.03 |
| Chitinase | 0.16 | 1.12 | 0.11 | −0.33 | −0.20 | −0.19 | −0.06 | −0.02 | 0.12 |
| Phosphatase | 1.27 | 1.19 | −0.26 | −0.25 | −0.22 | −0.31 | 0.00 | 0.02 | 0.02 |
| Cellulase | 1.03 | 0.77 | 0.01 | −0.26 | −0.23 | −0.28 | 0.00 | −0.02 | 0.16 |

[a]Relative changes are given for predicted gene abundances of Tax4Fun (Tax) and PanFP (Pan), as well as for measured enzyme activities in the rhizosphere of wheat and barley.

observed stronger positive correlations between the predictions and the measured activity for xylosidases and acid phosphatases in the wheat rhizosphere at maturity than at the active growth phase. In contrast, for barley, these correlations were stronger at the active growth phase than at crop maturity. Furthermore, the predictions by Tax4Fun at the active growth phase of barley, as well as the predictions by PanFP at both growth phases of barley were highly positively correlated with the measured phosphatase activities (Table S1).

Relative differences between active and mature growth phases, conventional and organic farming, and ambient and future climate for predicted and measured values are presented in Table 2. Thereby, relative differences of predicted functional gene abundances mostly mirrored the measured enzyme activities in a qualitative way, i.e., in terms of the direction (positive, negative, or no difference). Exceptions were found for acid phosphatases and, to a lower extent, also for glucosidases in the rhizosphere of barley. While predictions indicated higher gene abundances of the respective functional genes at the active phase than at the mature growth phase, measured enzyme activities showed an opposing pattern (Table 2).

**(ii) Concordance and discordance between the measured and predicted values.**
To be able to quantitatively compare the patterns of predicted gene abundances and measured enzyme activities, z-transformed data were used.

The degrees of over- or underestimations of functions varied between the tested experimental conditions (Fig. 3 and Fig. S1). Regarding the wheat rhizosphere, measured enzyme activities were mostly underestimated by the predictions at the active growth phase (Fig. S1), whereby the strongest deviations were found for conventional farming (CF) under ambient-climate conditions (Fig. 3) (Tax4Fun, standard deviation [SD] = 0.51; PanFP, SD = 0.63) and organic farming (OF) under future climatic conditions (Fig. 3) (Tax4Fun, SD = 0.65; PanFP, SD = 0.54). At crop maturity, we found strong concordances between activities and predictions (Fig. S1). They were particularly strong in OF under ambient-climate conditions (Fig. 3) (SD < 0.20 for both prediction tools) as well as in CF in an ambient climate (Fig. 3) (Tax4Fun, SD = 0.23; PanFP, SD = 0.36). Under future-climate conditions, the activities were underestimated in CF and overestimated in OF (Fig. 3). Overall, we found a better fit between measured activities and predictions in OF than in CF (Fig. S1) and an overall good fit for future- and ambient-climate treatments (Fig. S1).

For the barley rhizosphere, a clear pattern emerged with almost perfect fits of measured activities and predicted gene abundances at the active growth phase (Fig. 3) (SD < 0.2 for both prediction tools) (Fig. S1, red lines indicate zero deviations). The strongest deviations occurred in CF under future-climate conditions, when mainly phosphatase

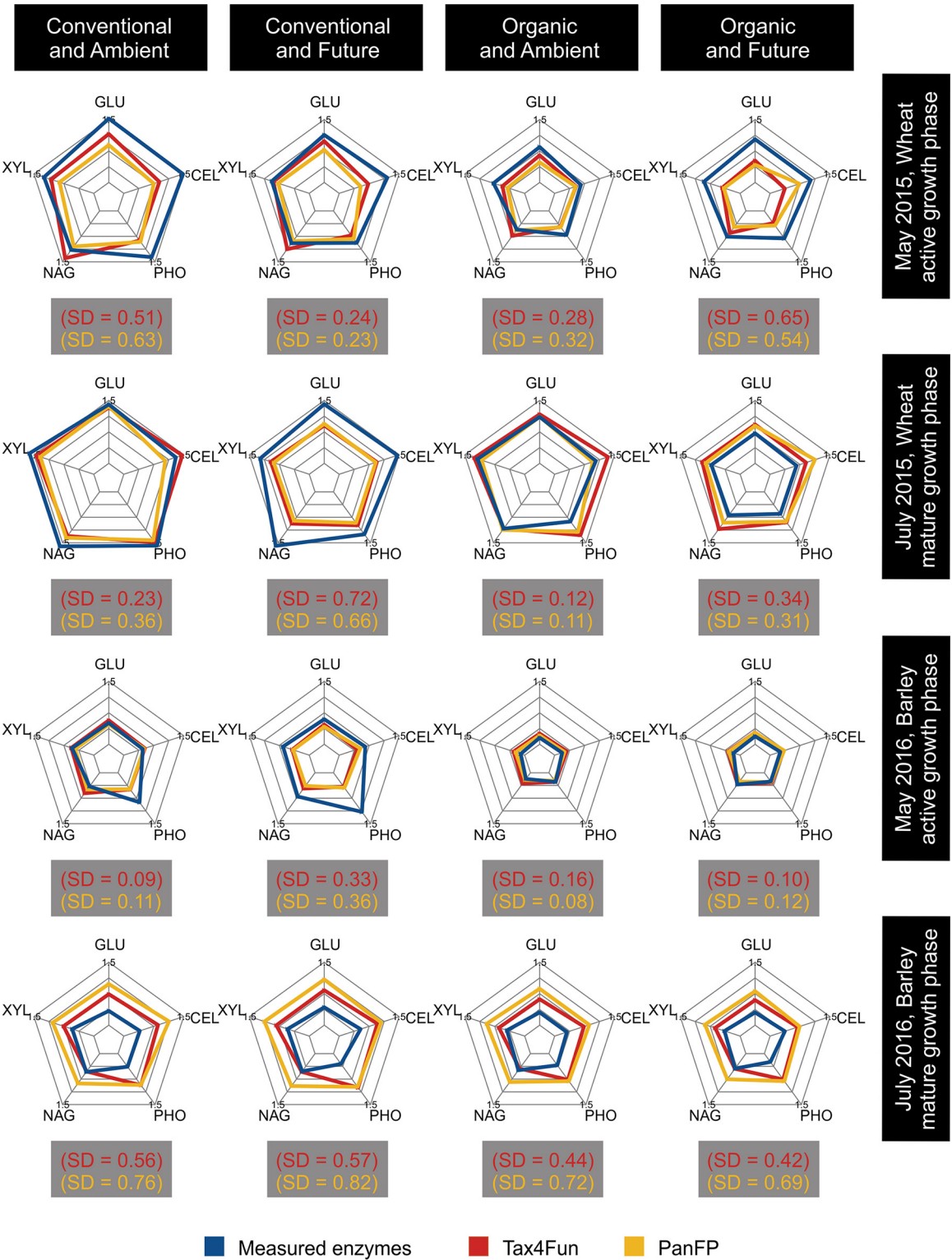

**FIG 3** Measured enzyme activities and predicted functional gene abundances of Tax4Fun and PanFP, arranged by growth phases of the two crops (horizontal) and by the experimental treatment (vertical). Data were normalized by z-transformation. The spider charts represent the measured enzyme activity levels (blue), and gene abundance levels estimated by Tax4Fun (red) and PanFP (yellow). Higher values are more distant from the center of the web. The median of the standard deviations between measured and predicted values for all enzymes is given for both prediction tools separately in brackets (SD). GLU, $\beta$-glucosidases; XYL, xylosidases; NAG, $N$-acetylglucosaminidases (chitinases); PHO, acid phosphatases; CEL, cellobiohydrolases (cellulases).

**TABLE 3** Significance of deviations in measured to predicted values with respect to the experimental factors[a]

| | P value[b] | | | | | | | | |
| | Growth phase | | | Farming system | | | Climate | | |
| Enzyme | Both | Wheat | Barley | Both | Wheat | Barley | Both | Wheat | Barley |
|---|---|---|---|---|---|---|---|---|---|
| **Tax4Fun vs enzymes** | | | | | | | | | |
| Glucose | *0.01* | 0.25 | *<0.001* | 0.44 | 0.41 | 0.87 | 0.52 | 0.63 | 0.52 |
| Xylosidase | 0.06 | 0.31 | *0.02* | 0.39 | 0.71 | 0.20 | 0.79 | 0.96 | 0.58 |
| Chitinase | 0.21 | 0.24 | 0.67 | 0.22 | 0.25 | 0.72 | 0.40 | 0.73 | 0.13 |
| Phosphatase | *<0.001* | *0.04* | *<0.001* | 0.05 | 0.21 | 0.05 | 0.71 | 0.73 | 0.84 |
| Cellulase | *<0.001* | *0.02* | *<0.001* | 0.15 | 0.16 | 0.54 | 0.10 | 0.18 | 0.20 |
| **PanFP vs enzymes** | | | | | | | | | |
| Glucose | *<0.001* | *0.03* | *<0.001* | 0.40 | 0.34 | 0.76 | 0.71 | 0.77 | 0.70 |
| Xylosidase | *0.003* | 0.26 | *<0.001* | 0.39 | 0.64 | 0.28 | 0.87 | 0.72 | 0.81 |
| Chitinase | 0.18 | 0.72 | *<0.001* | 0.24 | 0.20 | 0.88 | 0.32 | 0.52 | 0.27 |
| Phosphatase | *<0.001* | 0.15 | *<0.001* | 0.06 | 0.24 | 0.05 | 0.90 | 0.92 | 0.92 |
| Cellulase | *0.001* | 0.05 | *<0.001* | 0.05 | *0.003* | 0.90 | 0.74 | 0.80 | 0.37 |

[a]The deviation between z-transformed values of measured and predicted indices for the activity of five enzymes was calculated and tested for significance across growth phases, farming systems, and climate treatments.
[b]The P values are given according to ANOVA for the total data set (both wheat and barley, $n = 80$) as well as separately for wheat ($n = 40$) and barley ($n = 40$). Significant impacts according to ANOVA are indicated by italic font.

activities were heavily underestimated by the predictions (Tax4Fun, SD = 0.33; PanFP, SD = 0.36). At the mature growth phase of barley, enzyme activities were overestimated by the predictions (Fig. 3 and Fig. S1), except for chitinases, where Tax4Fun predictions perfectly matched measured activities (Fig. 3). Overall, we observed less deviations of the predicted from the measured values in OF than in CF (Fig. S1), while the deviations were similar across the two climate treatments (Fig. S1).

Testing the effect of different treatments on deviations, growth phase definitely exerted the strongest impact on the concordance between predicted and measured enzyme indices for individual enzymes in the wheat rhizosphere and for all of them in the barley rhizosphere (Table 3). Thereby, variance partitioning revealed that 22% of the total variations in the deviations can be explained by this treatment. In contrast, the experimental factors farming system and climate did not significantly affect the accuracy of the predictions (Table 3) (analysis of variance [ANOVA], $P > 0.05$; variance partitioning, 0.9% and 0.1%), and P values were smaller for farming system than for climate (Table 3). The decreasing differences in deviations of predictions from measured values, growth phase > farming system > climate treatment, was more obvious in the barley than in the wheat rhizosphere (Table 3).

## DISCUSSION

**Structure and function of microbial communities in agroecosystems are affected by different drivers.** Plant growth phase was the most prominent driver of community composition in the rhizospheres of wheat and barley. This result agrees with findings of Houlden et al. (48) and Francioli et al. (3), who demonstrated a strong shift in rhizobacterial community composition according to plant growth stages in crop plants such as wheat, pea, and sugar beet (48) but also in grassland species (3). Such differences can be explained by quantitative and qualitative changes in rhizodeposition that are related to the different plant development phases (46, 49).

In contrast, measured enzyme activities were mainly driven by farming practice and associated differences in mineral nitrogen and total C and N contents (see results in Material S1 in the supplemental material). Farming practice is an important driver of soil enzyme activity, which drastically changes soil structure (50, 51) and soil chemical parameters (25). We found higher enzyme activities in CF than in OF. In line with that, Arcand and colleagues (52) observed increased activity and production of enzymes in conventional farming soil compared to that in organic farming soil. The pattern may be caused by a higher availability of nitrogen in CF, which is known to foster the production and activity of polysaccharide-degrading enzymes (53–55).

Our results revealed that shifts in composition and functions of the rhizobacterial community are caused by different drivers, indicating a decoupling of community composition and function. In concordance, Francioli et al. (56) found that mineral and organic fertilizers mainly affect either activity or composition of the microbiome in an agricultural soil. Additionally, Bowles et al. (57) indicated that structurally highly similar bacterial communities can show very contrasting enzyme activities in differently managed organic fields.

**Measured enzyme activities partly confirmed by predictions.** The predicted gene abundances in our study responded to the drivers of both community composition and activity and were thus affected by the growth stage and by the farming system (Table S2, Fig. S2). In line with the first hypothesis, the performance of the prediction tools was driver dependent, whereby the strongest deviations could be related to the crop growth phase, followed by farming system and climate treatment. Furthermore, a remarkable impact of the crop species on the level of concordance between the predicted and measured activities was indicated.

For barley, in accordance with hypothesis two, strong correlations and concordances between measured activities and predicted gene abundances of both tools were observed at the active growth phase. In contrast, the enzyme activities were overestimated by both prediction tools at the mature growth phase. This finding is likely based on a faster response of the community composition than of enzyme activity. To promote plant growth, plants exudate carbon compounds into their rhizosphere that stimulate growth and activity of soil microorganisms (58 and reviewed in reference 59). Depending on plant development stage, root exudation patterns differ and thus strongly influence the rhizobacterial community (9, 46). Root exudation of carbon-rich compounds (sugar) is at its strongest in the juvenile growth phase, represented by the active growth phase of this experiment, and decreases thereafter (46). The overestimation of activity by the prediction tools at the mature growth phase of barley may therefore be a result of accumulated, mostly inactive rhizobacterial genes.

Deviation patterns of measured and predicted activities were more heterogeneous in the wheat rhizosphere. Three possible reasons may explain the discordance between predictions and measured activities:

(i) A major impact on enzyme activities was attributed to soil mineral N concentrations, which were positively correlated (Material S2) and significantly higher in the rhizosphere of wheat than in that of barley (Material S1). This relation has already been demonstrated for activities of cellulase and $\beta$-glucosidase (54, 60) as well as of acid phosphatase (61) and chitinase (62). The high mineral N concentration in the wheat rhizosphere likely fostered enzyme production without microbial growth, especially at the early active growth phase, and may explain deviations between measured activities and predictions.

(ii) For wheat, we observed an interaction effect of the experimental climate and farming system treatment on enzyme activities, which was not reflected in the predictions. Supporting our fourth hypothesis, we observed strong deviations under a future-climate condition at the mature growth phase and under organic farming conditions also at the active growth phase of wheat. With conventional farming at the active growth phase of wheat, the pattern of deviations was inverse. As such a pattern was not observed in the barley rhizosphere, this may indicate a plant-specific drought effect on the enzyme activity that could not be mirrored by the prediction tools. When Kosová et al. (63) summarized the knowledge about wheat and barley responses to drought, they stated no strong advantages or disadvantages for either of the two but variations along different genotypes. It is therefore likely that the cultivated wheat genotype expressed a different adaption capacity to drought than the barley genotype, which may feedback to structure and functionality of the rhizosphere community and thus functional predictions.

(iii) Indicator species analysis and a bipartite network indicated a clear effect of the growth phase for barley, which was also the main driver for community composition (Material S3). In contrast, the impact of wheat growth phase on indicator species was

surpassed by farming system, as we found high numbers of shared indicator species between active and mature growth phases under conventional farming conditions. Thus, the contradicting drivers for overall community and indicator species in the wheat rhizosphere may contribute to deviations between predictions and measured activities.

Interestingly, and against our assumptions of hypothesis three, farming system-related deviations between measured and predicted values were not obvious either in the wheat or in the barley rhizosphere. An explanation could be the normalization of the predicted functional gene abundances with 16S rRNA gene abundances, which were strongly biased by farming system. Another possibility would be that the effect of farming system was outcompeted by the overall stronger effect of growth phases. Besides the deviations between measured and predicted values, the correlations along the treatments and relative changes between factors of treatments were not affected, suggesting that the prediction tools mirrored the impacts of the experimental factors in a qualitative way.

**Limitations. (i) Prediction tools.** Although the applied tools were created for universal use, their predictive power depends on the quality of the databases. Another commonly used prediction tool, PiCrust (phylogenetic investigation of communities by reconstruction of unobserved states [64]), is tailored to functional predictions in the human microbiome by using the Integrated Microbial Genomes database (65) containing genomes from the Human Microbiome Project (The Human Microbiome Jumpstart Reference Strains Consortium [66]) and Greengenes database (67). For soil microbiomes, Tax4Fun (36) and PanFP (37) outperform predictions of PiCrust (36, 37). These tools rely on the bacterial sequences of the SILVA database (68) which comprises quality-controlled aligned rRNA gene sequences. Nevertheless, all tools are subject to some restrictions which have to be considered for analysis.

First, the contribution of other organisms, including plants and fungi, to the extracellular enzyme production cannot be estimated by the tools. While extracellular enzymes in the soil are mainly attributed to origination from edaphic microorganisms and the contribution of plants may be thus negligible (39, 69), the contribution of fungal communities is of considerable importance, especially in extensively managed grassland and forest ecosystems (70). For comparison of predicted and measured activities, we therefore performed our study in agricultural systems which are known to be dominated by bacterial communities (41, 42).

Second, the predictive power of the tools relies on the integrity of the databases. SILVA (68) and KEGG Orthology (KO) (71) databases facilitate annotation of bacteria preferably to the genus level (36), thereby loosing information about functional differences on species level. Aßhauer et al. (36) further noticed that the members of the highly diverse soil community are not sufficiently represented in the KEGG database. Since publication of the prediction tools in 2015, the SILVA and KO databases have been updated frequently. Notwithstanding, the tools do not implement the latest versions of KO and SILVA databases. While PanFP is based on SILVA v128 (released in 2017), Tax4Fun supports only SILVA v123 (released in 2015). The KO database has been licensed and, thus, allows only free access to version 64.0 released in October 2012. In our analysis, we used the latest applicable versions for Tax4Fun and PanFP analysis. For Tax4Fun predictions, we were able to trace back the percentage of bacterial sequences used for predictions and the distribution of KO identifiers representing the five enzymatic classes among samples (Fig. S3). Only 37% and 32% of bacterial sequences could be used for the functional predictions in the rhizospheres of wheat and barley, respectively. Regarding indicator species, the numbers improved to at least 43% for wheat and 42% for barley. It would, therefore, be desirable to integrate updated databases and more recent metagenomic data to improve predictions.

Third, the expected discrepancy between gene presence and expression is a major constraint. Next-generation sequencing does not discriminate between the fractions of living and dead cells. Furthermore, in a given soil, only a certain proportion of

microorganisms are active at a certain time point (reviewed in references 72). The identity and number of active or inactive taxa depend on external conditions and stimuli (73). In our study, we specifically investigated the impact of land use/management, climate, and plant growth. Our results imply that the various proportions of active taxa and the contributions of dead organisms may partly explain discrepancy between measured and predicted activities and may be the main reason for the missing quantitative concordance.

**(ii) Evaluation of enzyme activities as functional indicators.** Besides database-related deficiencies, discrepancies in the comparisons may also emerge due to the approach used to determine enzyme activities. The measurement of soil enzyme activities cannot distinguish between recently secreted enzymes and enzymes that were produced earlier by taxa whose relative abundance may have declined. These so-called abiotic enzymes are protected against their degradation in clay complexes (39). The persistence and accumulation of enzymes in the soil matrix may strongly influence overall enzyme activity measurement. The amount of immobilized enzymes strongly depends on the soil type and respective clay and organic matter content (74). We found a strong relationship between 16S rRNA gene abundances and enzyme activities at the active growth phase of both crops but a decrease of enzyme activity accompanied by an increase of gene abundances in the mature growth phase. This indicates a minor role of abiotic enzymes for the measured activities and, rather, suggests different root exudation rates at active and mature growth phases as the main driver of enzyme secretion.

A second methodological aspect that should be considered is that we determined enzyme activities under standardized conditions which are close to the optima of the different enzymes (pH 5, 25°C). Moreover, the high substrate concentration in the assay (300 $\mu$M) ensured no limitation due to substrate availability (75). While the used temperature (25°C) represents reasonable daytime temperatures at the sampling dates in late May and July, the used pH was much lower than the average from all samples (pH 6.3). Since we did not compare absolute values but only differential expression of enzymes, and the pH was comparable between all treatments, differences along the treatments should be maintained. In contrast, farming system-specific substrate availability may result in different enzyme patterns than those obtained under saturating substrate concentrations. These methodological issues plead for further studies measuring enzyme activities at realistic temperature and pH as well as particularly under substrate concentrations reflecting the availability in the respective system.

**Conclusions and perspectives.** Our results demonstrate that Tax4Fun and PanFP provide cost-effective tools to estimate functional patterns of rhizobacterial communities in a qualitative (i.e., direction of response) but not in a quantitative (i.e., extent of response) way. The response of the studied activities to experimental drivers was predominantly predicted correctly by both tools in terms of direction, i.e., increase or decrease. This is particularly noteworthy, since drivers of community structure and measured activities differed from each other. Moreover, we observed a gradual decrease in predictability the faster the treatments acted on community structure. The tools do not provide a one-size-fits-all solution, and interpretation of predicted functions has to be performed thoroughly. To trace mechanisms behind concordances and discordances, a deeper understanding of the underlying drivers of functions and structure is necessary. We emphasize the importance of more studies on predicted and measured functional traits to explore relevant drivers for functions and structure in different environments. These studies should be combined with transcriptomics data to explore whether the link between predicted and real soil processes will be strengthened, as assumed for this study, or, rather, diminished.

Nevertheless, our finding provides a valuable validation of the applicability and robustness of these prediction tools for comparison of soil community functions across stable soil ecosystems. While the enzymes used in this approach address solely activities related to nutrient cycling, the positive validation would plead for further research

on the possibility to predict other, more intractable functions of microbial communities in a simple and quick way.

## MATERIALS AND METHODS

**Soil sampling and sample preparation.** Samples were obtained from the Global Change Experimental Facility (GCEF) situated at the research station in Bad Lauchstädt, Central Germany (51°23′35′′N 11°52′55′′E, 118 m above sea level [a.s.l.]). The site is characterized by a temperate climate with an average temperature of 9.7°C (1993 to 2013) and a mean annual precipitation of 525 mm (1993 to 2013). The soil type is a fertile loamy soil (haplic chernozem) (76). The experimental platform of the GCEF was established in 2013 and combines land use and climate treatments as described by Schädler et al. (44). In our study, we focused on the cultivated cropland systems, organic farming (OF; 10 plots) and conventional farming (CF; 10 plots). The grown crop was identical for CF and OF in 2015 (winter wheat) and 2016 (winter barley) but differed in 2014 (CF, winter rape seed; OF, field bean). In CF, synthetic fertilizers (N, P, and K), growth regulators, and pesticides are applied. The use of pesticides in OF is restricted, and fertilization is realized by including legumes in the crop rotation as well as by the application of rock phosphate (P-Ca-Mg) and patent kali (K-Mg-S) every 3 years. Half of the plots experience ambient climate (A), while the other half is exerted to simulated future-climate conditions (F) comprising a warming ($+0.55$°C on average) and a changed precipitation pattern ($-20\%$ in summer, $+10\%$ in spring and fall) (for details refer to Schädler et al. [44]). Cereal roots with closely adherent soil were sampled in the active cereal growth phase in May and at the mature state in July in 2015 (3 wheat plants per plot, 20 plots) and 2016 (3 barley plants, 20 plots). Samples were transported in cooling boxes to the field station and immediately frozen. Simultaneous to root sampling, surrounding bulk soil was sampled for the analysis of soil parameters. For this, six soil cores (diameter [Ø] 15 mm, 0- to 15-cm depth) were taken from each plot, pooled, sieved to 2 mm, manually cleaned from organic material, and frozen at $-20$°C.

To separate rhizosphere soil from roots, the roots with adherent soil were crushed and put in 50-ml Falcon tubes with 40 ml of 0.5% NaCl solution. Tubes were vortexed for 1 min to loosen adherent soil from the roots. Subsequently, roots were transferred to a second set of 50-ml Falcon tubes. Soil suspensions without roots were centrifuged at $12,851 \times g$ for 10 min. Then, the supernatants were filled into the tubes with the roots and used for a second washing step. After vortexing, soil suspensions were transferred into the Falcon tubes with the pellets from the first washing step. The procedure of washing and centrifuging was repeated three times. Rhizosphere soil pellets were frozen at $-20$°C.

**Soil parameters.** Since the amount of rhizosphere soil was limited to 2 to 3 g per sample, basic soil parameters were determined using respective bulk soil. For pH analysis, 12 g of air-dried soil was suspended in 30 ml of 0.01 M $CaCl_2$ solution (1:2.5 [wt/vol]). The soil suspension was equilibrated at room temperature and thoroughly mixed every 20 min. After 1 h, the pH was measured with a pH electrode. Total carbon and nitrogen contents were determined from air-dried soil using an elemental analyzer (Elementar Vario EL III; Elementar, Hanau, Germany). For analysis of mineral nitrogen, 5 g of fresh soil was suspended in 20 ml of 1 M KCl solution and measured via flow injection analysis (FIAstar 5000; Foss GmbH, Rellingen, Germany). Available phosphorus was extracted from fresh soil with double lactate (1:50 [wt/vol], pH 3.6) and quantified using the colorimetrical molybdenum blue method (77).

**Soil enzymes.** The activity potentials of hydrolytic soil enzymes were measured using a modified fluorometric assay introduced by Sinsabaugh et al. (78). The analyzed enzymes are involved in phosphorus acquisition (phosphatases), nitrogen acquisition (*N*-acetylglucosaminidases), and carbon acquisition ($\beta$-glucosidases, xylosidases, and cellobiohydrolases). Enzymatic activities were determined as turnover rate of 4-methylumbelliferon (MUF)-coupled substrates (Table S3 in the supplemental material), where the amount of released fluorescent MUF was directly related to enzymatic activity potentials. To avoid underestimation of enzyme activities (79), the substrate concentration was optimized for the haplic chernozem soil and set to 300 $\mu$M for all substrates.

For each sample, a black 96-well microplate was prepared containing all five substrates, MUF dilutions to calculate quench and extinction coefficients (1.25 $\mu$M and 2.5 $\mu$M), and the substrate and soil suspension controls. For analysis, approximately 250 mg of fresh rhizosphere soil was suspended in 50 ml of 50 $\mu$M acetate buffer (pH 5) and sonicated for 5 min to break up soil aggregates. Subsequently, the soil suspension was added to the substrates and incubated at 25°C for 60 min. The enzyme reaction was stopped by the addition of 1 M NaOH solution. After 3 min, fluorescence was measured for eight replicates using an Infinite 200 PRO instrument (Tecan Group Ltd., Männedorf, Switzerland) with 360-nm excitation and 465-nm emission filters. Enzyme activity was calculated as turnover rate of substrate in nanomoles per gram dry soil per hour (nmol g soil$^{-1}$ h$^{-1}$) (80).

**DNA extraction and next-generation sequencing (Illumina MiSeq).** The extraction of soil bacterial genomic DNA was performed using the PowerSoil DNA isolation kit (MO BIO Laboratories Inc., Carlsbad, CA, USA). The protocol was slightly modified by increasing the soil amount from 250 to 400 mg. The concentration of extracted DNA was examined with a NanoDrop ND-8000 spectrophotometer (Thermo Fischer Scientific, Dreieich, Germany), and the DNA was then stored at $-20$°C. Before running the PCR, the concentrations of DNA extracts were adjusted to 10 to 15 ng/$\mu$l. The amplification of the bacterial 16S rRNA gene V4 region was performed with the universal primers 515f and 806r (81), which were equipped with Illumina adapter sequences. To ensure correct amplification of the sequences, all PCRs were performed using proofreading KAPA HiFi polymerase (KAPA Biosystems, Boston, MA, USA). The conditions of the PCR are summarized in Table S4 (PCR 1).

PCR products were tested by gel electrophoresis and purified using the Agencourt AMPure XP kit (Beckmann Coulter, Krefeld, Germany). To assign the sequences to the respective samples, Illumina Nextera XT indices were attached to both ends of the bacterial fragments in a second PCR. The

conditions of the index PCR are presented in Table S4 (PCR 2). PCR products were purified using AMPure beads, and DNA was quantified with the PicoGreen assay (Molecular Probes, Eugene, OR, USA). For an equimolar representation of each sample, defined volumes of prepared bacterial amplicon libraries (corresponding to 80 ng DNA for each sample) were pooled in one tube. The fragment sizes and the quality of DNA sequencing libraries were again checked with the Agilent 2100 bioanalyzer (Agilent Technologies, Palo Alto, CA, USA). Sample libraries and PhiX control libraries were denatured according to the protocol of the MiSeq v3 reagent kit and diluted to a final concentration of 10 pM. Denatured and diluted libraries were combined to a volume of 600 $\mu$l (30 $\mu$l of PhiX control library, and 570 $\mu$l of bacterial amplicon library) and loaded onto MiSeq v3 reagent cartridge for sequencing. Finally, paired-end sequencing of 2 by 300 bp was implemented on an Illumina MiSeq platform (Illumina Inc., San Diego, CA, USA) at the Department of Soil Ecology of the Helmholtz Centre for Environmental Research (UFZ, Halle, Germany).

**Bioinformatics workflow, functional predictions, and normalization.** In total, 13,912,979 demultiplexed sequencing reads were processed using an in-house pipeline (described in reference 82, with modifications) based on mothur (83) and OBITools (84). In brief, reads without the 515f and 806r primers were discarded, and the primers were clipped from the remaining sequences using cutadapt (85). Read pairs were assembled using PANDAseq (86) and quality trimmed to an average Phred score of 26, retaining 59% of the reads. After preclustering at 99% identity using CD-HIT-454 (87), chimeric reads were removed using the UCHIME algorithm (88), and the remaining reads were clustered into operational taxonomic units (OTUs) at 97% identity using vsearch (89). To safeguard against artifacts, singletons were removed corresponding to 2.5% of the reads. The representative reads of all OTUs were examined for chimeric reads, which were removed using the UCHIME algorithm. Subsequently, the representative reads were taxonomically assigned based on the reference sequences from the SILVA database (version 128, nonredundant at 99% [90]) using the mothur implementation of the naive Bayesian classifier (91). OTUs of chloroplasts and mitochondria and those not assigned to the kingdoms *Bacteria* or *Archaea* were removed. The final sequencing depth per sample ($\pm$ standard deviation) was 96,000 $\pm$ 14,000 reads.

Functional predictions of the bacterial communities were performed with two programs, PanFP (37) and Tax4Fun (36), working on the basis of the OTU abundance and taxonomy data. Tax4Fun (36) and PanFP (37) create functional profiles of bacterial communities using two related approaches for the analyses. Tax4Fun assigns operational taxonomic units (OTUs) to reference sequences in the SILVA database (SILVA database [68]) and converts the counts of SILVA-labeled OTUs to a taxonomic profile of organisms in the KEGG database (71). PanFP creates pangenomes of taxonomically related genomes with their identity also obtained via the SILVA database and subsequently weighs the pangenome's functional profile with OTU abundances. The authors' instructions were followed to run Tax4Fun program line in R (version 3.4.0 [92]) using the latest supported version of SILVA database (SILVA123, released July 2015). Taxonomic assignment was adapted for Tax4Fun analysis to SILVA123. PanFP was executed on a suitably formatted OTU table. The output tables provided KEGG orthology (KO) numbers for gene annotations and Enzyme Commission (EC) number as object identifier for enzymes. Gene abundances of the enzymes of interest were extracted from the output tables of Tax4Fun and PanFP predictions (Table S5). Gene abundances of the three acid phosphatases and the two $\beta$-glucosidases, which belong to the same enzymatic class according to EC numbers, were summed up from the Tax4Fun and PanFP output tables, respectively, for further analysis.

Both programs provide abundance estimates of functional genes which are compromised by methodical restraints of 16S rRNA gene sequencing. To allow a balanced reading, the input samples had to be adjusted to a certain DNA concentration, vanishing actual differences between samples. To correct predictions of gene abundances for biomass differences in the samples (36, 37), we estimated bacterial DNA concentrations of rhizosphere samples by quantitative real-time PCR (qPCR) analysis. In conformity with the Illumina sequencing, the reactions were performed with the primer pair 515f and 806r (81) targeting the 16S rRNA gene V4 region of the bacterial genomes. All samples were diluted to 2-ng/$\mu$l DNA input concentrations, as measured by the PicoGreen assay (Molecular Probes, Eugene, OR, USA), and the dilution factor was recorded. As reference, a dilution series of 0.05, 0.125, 0.5, 1, and 4 ng/$\mu$l genomic DNA of a *Phyllobacterium* isolate from the same soil was prepared to generate a standard curve. Quantitative PCR was run with the Bio-Rad iCycler (Bio-Rad Laboratories GmbH, Munich, Germany) under the conditions listed in Table S4 (qPCR). Measured threshold cycle ($C_T$) values of the rhizosphere samples were related to the standard curve to calculate the mass of bacterial DNA. Subsequently, the obtained DNA concentrations were multiplied with the dilution factors to yield the relative bacterial DNA concentrations for each sample. The obtained values were used as factors to normalize gene abundances in each sample and are further given as DNA concentrations in micrograms per gram dry soil in Table S6.

**Statistics.** All analyses were performed with the open-source software R (version 3.4.0, R Core Team). The impacts of farming system, climate, and growth phases were tested separately for OTU data to identify the main drivers of the bacterial community. The factors were then ordered by decreasing impact. PERMANOVA ("adonis" R package vegan) was run, separately for crop species, using the following model: Bray-Curtis dissimilarity (OTU table) $\approx$ growth phase $\times$ farming system $\times$ climate. PERMANOVA ("adonis," R package vegan) was also performed to analyze the influence of soil abiotic parameters on community structure. The stratification by crop species ensured permutations only within groups of samples belonging to wheat or barley rhizosphere. To visualize significant grouping factors of bacterial community composition, principal-coordinate analysis (PCoA) was performed. For this, absolute abundances of each OTU were normalized to the total read counts in the samples, a Bray-Curtis dissimilarity matrix was calculated, and the first two axes of the PCoA were plotted.

Indicator species analysis was performed to identify OTUs that were either specific or shared between wheat and barley among the two growth stages and the two farming systems. According to Hartman et al. (93), we applied two different approaches to test for indicator species using R (version 3.4.0 [92]). The correlation-based approach calculates point-biserial correlation coefficients (R package indicspecies [94]) indicating positive associations of OTUs to one or various conditions of farming system and plant growth phases. Associations were considered significant at a $P$ value of <0.05. A likelihood ratio test evaluated differences in abundances of OTUs between plant growth phases in the two different farming systems (R package edgeR [95]). Differences in abundances were considered significant by a false-discovery rate (FDR)-corrected $P$ value of <0.05. OTUs, when confirmed by both tests to be significant, were regarded as indicator species and further implemented in bipartite network analysis. The network was constructed with the R package igraph (93, 96).

Subsequently, Fisher tests were performed identifying enrichments of phyla within the indicator species compared to the overall community composition in wheat and barley rhizospheres to examine if indicator species were a random subset of the overall community (R package rcompanion [97]).

The measured enzyme activities of glucosidases, xylosidases, chitinases, phosphatases, and cellulases were evaluated using the following linear model: test variable $\approx$ (crop species) $\times$ farming system $\times$ growth phase $\times$ climate.

The crop species was included when analyzing effects across both years/crop species. To test for significant impacts of single factors and for interaction effects, but also for the influence of abiotic soil parameters, an ANOVA was run with the respective models followed by Tukey's honestly significant difference (HSD) *post hoc* tests. Significance levels were classified as highly significant ($P < 0.001$), strongly significant ($P < 0.01$), significant ($P < 0.05$), and tendency ($P < 0.1$.).

Spearman rank correlation tests were applied to assess correlations of measured enzyme activities with the respective predicted gene abundances within climate and farming system treatments and also within and among the different growth phases. To compare expression levels of measured and predicted enzyme activities, values were z-transformed. The differences between measured and predicted Tax4Fun and PanFP, as well as relative changes between factors of the treatment, growth phase, farming system, and climate, were calculated for each enzyme. Significant deviations within and among experimental factors tested with ANOVA and variance partitioning are given (R package *vegan*).

**Data availability** Demultiplexed sequences are accessible in the Sequence Read Archive under BioProject accession PRJNA605022. The pipeline used for analysis of metabarcoding raw read libraries is available at https://github.com/lentendu/DeltaMP. Tax4Fun and PanFP are open access, author's descriptions can be found for Tax4Fun under http://tax4fun.gobics.de/ and for PanFP under https://github.com/srjun/PanFP.

## SUPPLEMENTAL MATERIAL

Supplemental material is available online only.

**SUPPLEMENTAL FILE 1**, PDF file, 2.1 MB.

## ACKNOWLEDGMENTS

We thank the Helmholtz Association, the Federal Ministry of Education and Research, the State Ministry of Science and Economy of Saxony-Anhalt, and the State Ministry for Higher Education, Research and the Arts Saxony to fund the Global Change Experimental Facility (GCEF) project. We also thank the staff of the Bad Lauchstädt Experimental Research Station (especially Ines Merbach and Konrad Kirsch) and Martin Schädler for their work in maintaining the plots and infrastructures of the GCEF as well as Harald Auge, Stefan Klotz, and Martin Schädler for their role in setting up the GCEF. We thank Beatrix Schnabel for performing the sequencing. We thank Guillaume Lentendu for establishing the sequence processing workflow. The community composition data were computed at the High-Performance Computing (HPC) Cluster EVE, a joint effort of both the Helmholtz Centre for Environmental Research—UFZ and the German Centre for Integrative Biodiversity Research (iDiv) Halle-Jena-Leipzig.

Claudia Breitkreuz was supported by funding from the Deutsche Bundesstiftung Umwelt (DBU; AZ 20015/391) and sequencing costs by the Helmholtz Centre for Environmental Research—UFZ. Anna Heintz-Buschart was funded by the German Centre for Integrative Biodiversity Research (iDiv) Halle-Jena-Leipzig of the German Research Foundation (FZT 118, 202548816). Sara Fareed Mohamed Wahdan is financially supported by the Egyptian Scholarship (Ministry of Higher Education, external missions 2016/2017 call).

F.B., M.T., T.R., A.H.-B., and C.B. conceived and designed the experiment. T.R. and C.B. performed the field experiments. C.B., T.R., and S.F.M.W. performed the laboratory work. A.H.-B. ran bioinformatics. C.B. and A.H.-B. analyzed data. Results were interpreted by

M.T., T.R., F.B., A.H.-B., and C.B. M.T., T.R., A.H.-B., and C.B. wrote the manuscript with input from F.B. and S.F.M.W.

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
