## [Reviewer comments · Microbiology Spectrum]

**Microbiology
Spectrum**

Can we estimate functionality of soil microbial communities from structure-derived predictions? A reality test in agricultural soils

Claudia Breitzkreuz, Anna Heintz-Buschart, Francois Buscot, Sara Wahdan, Mika Tarkka, and Thomas Reitz

Corresponding Author(s): Claudia Breitzkreuz, Helmholtz Centre for Environmental Research

Review Timeline:

Submission Date:

June 29, 2021

Accepted:

July 12, 2021

Editor: Jeffrey Gralnick

Reviewer(s): The reviewers have opted to remain anonymous.

Transaction Report:

DOI: <https://doi.org/10.1128/Spectrum.00278-21>

July 12, 2021

Dr. Claudia Breikreuz
Helmholtz Centre for Environmental Research
Soil Ecology
Theodor-Lieser-Strasse 4
Halle, Saxony Anhalt 06120
Germany

Re: Spectrum00278-21 (Can we estimate functionality of soil microbial communities from structure-derived predictions? A reality test in agricultural soils)

Dear Dr. Claudia Breikreuz:

Based on your thorough responses to the AEM reviewers and in consultation with another Editor here at Spectrum, your manuscript has been accepted, and I am forwarding it to the ASM Journals Department for publication. You will be notified when your proofs are ready to be viewed.

Sincerely,

Jeffrey Gralnick
Editor, Microbiology Spectrum
